# Apoptosis, Cell Growth, and Glycogen Synthase Kinase 3β Phosphorylation in Caffeic Acid-Treated Human Malignant Melanoma Cells

**DOI:** 10.3390/biomedicines13102389

**Published:** 2025-09-29

**Authors:** Yoon-Jin Lee, Ki Dam Kim, Min Hyuk Choi, Sukh Que Park, Yu Sung Choi, Youin Bae, Hae Seon Nam, Sang Han Lee, Moon Kyun Cho

**Affiliations:** 1Department of Biochemistry, Soonchunhyang University College of Medicine, Cheonan 31151, Republic of Korea; leeyj@sch.ac.kr (Y.-J.L.); m1037624@sch.ac.kr (S.H.L.); 2Division of Molecular Cancer Research, Soonchunhyang Medical Research Institute, Soonchunhyang University, Cheonan 31151, Republic of Korea; 19880016@schmc.ac.kr; 3Department of Dermatology, Soonchunhyang University Hospital, Seoul 04401, Republic of Korea; 132351@schmc.ac.kr (K.D.K.); 146079@schmc.ac.kr (M.H.C.); 73301@schmc.ac.kr (Y.S.C.); 84054@schmc.ac.kr (Y.B.); 4Department of Neurosurgery, Soonchunhyang University Hospital, Seoul 04401, Republic of Korea; drcolor@schmc.ac.kr

**Keywords:** caffeic acid, melanoma, GSK3β signaling, apoptosis

## Abstract

**Objectives:** Caffeic acid (CA), a naturally occurring phenolic compound exhibiting antioxidant and anti-inflammatory effects, has demonstrated anticancer activity against several tumor types. Nevertheless, its involvement in melanoma and its effects on the GSK3β signaling pathway have not been fully elucidated. This study aimed to assess the expression of p-GSK3β in melanoma tissues and to evaluate the anti-melanoma efficacy of CA. **Methods:** Western blot analysis was performed to determine the expression levels of p-GSK3β in melanoma and normal skin samples. G361 melanoma cells were exposed to CA, after which cell viability, apoptotic induction, cell cycle distribution, and related signaling molecules were examined. **Results:** Significantly increased p-GSK3β levels were identified in melanoma tissues. CA exposure decreased cell viability, triggered apoptosis, and elevated p-GSK3β levels in G361 melanoma cells. Moreover, CA induced the upregulation of p53 and p21 while concomitantly downregulating cyclin D1 and Bcl-2. **Conclusions:** These results suggest that CA inhibits melanoma cell growth through activation of a pathway involving the tumor suppressor p53, rather than through modulation of GSK3β signaling.

## 1. Introduction

Melanoma represents the most fatal form of skin malignancy, originating from pigment-producing melanocytes [1]. Although melanoma comprises only a minor fraction of all skin cancers, it is responsible for the majority of deaths attributable to skin cancer [1,2]. The poor prognosis associated with melanoma largely results from its marked capacity to infiltrate dermal tissues beyond the epidermis and to metastasize rapidly to distant sites, including lymph nodes, lungs, liver, and brain [3]. A wide range of cell signaling networks contribute to the initiation and progression of melanoma [4].

Glycogen synthase kinase 3β (GSK3β), a multifunctional serine/threonine protein kinase, serves as a central enzyme regulating diverse cellular processes, such as cell signaling, proliferation, differentiation, apoptosis, and the cell cycle [5,6]. GSK3β influences cancer cell proliferation largely through its modulation of oncogenic pathways, most notably the Wnt/β-catenin signaling pathway, wherein it phosphorylates β-catenin, resulting in its proteasomal degradation and subsequent inhibition of cell proliferation [7,8]. The phosphorylation of GSK3β at Ser9 represents a critical regulatory mechanism that inhibits the normal enzymatic activity of GSK3β [5,9]. LiCl is a well-established inhibitor of GSK3β, and its administration has been demonstrated to enhance Ser9 phosphorylation across various tissues and cell types, transitioning GSK3β into a phosphorylated, inactive form (p-GSK3β) [10].

Although GSK3β is abundantly expressed in a range of human cancers, its specific function as a tumor promoter or suppressor is still under debate [9,11]. In addition, multiple investigations have documented elevated levels of the inactive p-GSK3β form in several cancer types [12,13]. A summary of GSK3β dysregulation across different cancer types is provided in Table 1 [12,13,14,15,16,17,18]. Nonetheless, even though GSK3β’s involvement in cancer biology is well recognized, there is a notable paucity of studies addressing GSK3β expression specifically in human melanoma tissues. Additionally, insufficient research exists regarding p-GSK3β expression levels in melanoma and the contribution of p-GSK3β to the initiation and progression of this malignancy.

Caffeic acid (CA), a naturally occurring phenolic compound in the hydroxycinnamic acid family, is predominantly derived from honeybee propolis and is present in a variety of plants, including coffee beans, fruits, and vegetables [19,20]. Increasing research indicates that CA possesses notable anticancer activity via several mechanisms, such as triggering apoptosis, causing cell cycle arrest, and inhibiting angiogenesis in a range of malignancies [21,22]. Reports have shown that CA suppresses cancer progression by inhibiting cell migration, invasion, and metastasis in cancers such as breast, prostate, and lung [23]. Despite the extensive evidence supporting CA’s anticancer effects across multiple cancer types, its precise therapeutic potential and molecular mechanisms underlying its effects in melanoma—one of the most aggressive forms of skin cancer—are not yet well understood and need thorough investigation.

Accordingly, this study was conducted to evaluate the expression of p-GSK3β in human melanoma tissues and to further elucidate the anticancer effects of CA on melanoma development. We specifically analyzed the effects of CA treatment on factors such as cell viability, cell cycle progression, apoptosis induction, and the modulation of p-GSK3β expression in melanoma cell lines, aiming to determine the potential of CA as a novel therapeutic strategy for melanoma.

## 2. Materials and Methods

### 2.1. Tissue Sample Collection and Preparation

A total of six normal skin tissues and six melanoma tissues were collected from patients undergoing surgery between December 2015 and November 2018 in the Department of Plastic and Reconstructive Surgery at Soonchunhyang University Bucheon Hospital, Korea. The use of these samples for research was reviewed and approved by the Institutional Review Board (IRB) of Seoul Soonchunhyang University Hospital (IRB File no. 2019-04-029). Six melanoma tissue samples were collected from three male and three female patients and were histologically confirmed by a pathologist. Normal skin tissues were obtained from the backs of six women who underwent breast reconstruction with a latissimus dorsi flap. Remaining tissue specimens were immediately frozen in liquid nitrogen after resection and stored at −70 °C for subsequent Western blot analysis.

### 2.2. Cell Culture

The human malignant melanoma cell line G361 and SK-MEL-24 was obtained from the American Type Culture Collection (Manassas, VA, USA). Cells were cultured in DMEM supplemented with 5% fetal bovine serum (FBS), 1 mM glutamine, 100 units/mL penicillin, and 100 µg/mL streptomycin at 37 °C in a 5% CO_2_ incubator.

### 2.3. Western Blot Analysis

Stored tissue samples were homogenized using a protein extraction buffer (Tris-HCl: 50 mM, pH 7.4, NP-40: 1%, Na-deoxycholate: 0.25%, NaCl: 150 mM, protease inhibitor cocktails (Roche, Complete, Mini 1836153)), and Western blot analysis was performed as described by Lee et al. [24] The total protein was extracted and quantified. The Bradford solution (Bio-Rad protein assay, #600-0005, Bio-Rad Laboratories, Hercules, CA, USA) was diluted to 20%. The protein concentrations of the samples were determined using an ELISA reader and a standard curve.

Extracted protein samples (50 µg/µL) were separated using a NuPAGE 4–12% Bis-Tris gel (Invitrogen NP0335Box, Thermo Fisher Scientific, Waltham, MA, USA). Following SDS-PAGE, proteins were transferred to a PVDF membrane (RPN303F, Cytiva, Amersham, UK), and the membrane was blocked with blocking solution (5% skim milk (BD 232100, Becton, Dickinson and Company, Franklin Lakes, NJ, USA), 10 mM Tris-Cl, pH 8.0, 150 mM NaCl, 0.05% Tween 20) for 2 h. The membrane was incubated with primary antibody p-GSK3β (Santa Cruz Biotechnology SC-3738, Santa Cruz, CA, USA) and diluted to 1:1000 at 4 °C for 16 h. The membranes were washed twice for 10 min and twice for 15 min with washing buffer and TBST buffer (10 mM Tris-Cl, pH 8.0, 150 mM NaCl, 0.05% Tween 20). Afterward, p-GSK3β was detected using Anti-goat IgG (Santa Cruz Biotechnology SC-2020, CA, USA), diluted to 1:3000 for 2 h. Following incubation with the secondary antibody, the membrane was washed twice for 10 min and twice for 15 min. Protein detection was accomplished with an enhanced chemiluminescence solution kit (Cytiva, Amersham, UK). The membranes were subsequently stripped and reprobed with beta-actin antibody (A5441; Sigma-Aldrich, St. Louis, MO, USA). The quantification of protein expression levels was performed using ImageJ software (Version 1.54k; National Institutes of Health, Bethesda, MD, USA), a JAVA-based image analysis program developed by the U.S. National Institutes of Health and available online.

### 2.4. Cell Viability Assays

We used the procedures outlined by Lee et al. [24]. After CA treatment for 24, 48, and 72 h, cell viability was assessed using the MTT assay. MTT (final concentration 0.1 mg/mL) was added to the culture medium, and cells were incubated for an additional 4 h. After removing the medium, formazan crystals generated by the reduction of MTT by mitochondrial dehydrogenases in viable cells were dissolved in 500 µL DMSO and quantified spectrophotometrically at 555 nm. The results were expressed as a percentage, calculated as the ratio of absorbance of treated cells to that of the controls (100%).

### 2.5. Measurement of ATP Content

Intracellular ATP concentrations were quantified using the CellTiter-Glo Luminescent Cell Viability Assay kit (Promega Corporation, Madison, WI, USA). Specifically, cells underwent CA treatment for 48 h, after which CellTiter-Glo reagent (100 μL/well) was added directly to the cultures, followed by shaking for 2 min, and incubation at room temperature (RT) for 10 min to achieve cell lysis. Luminescent signals were subsequently recorded on the GloMax-Multi Microplate Multimode reader (Promega Corporation). ATP levels were normalized relative to cell count, determined using a trypan blue exclusion assay for each treatment and time point.

### 2.6. 4′,6-Diamidino-2-Phenylindole (DAPI) Staining

Nuclear condensation and fragmentation were observed by nucleic acid staining with DAPI. Cells were treated with fisetin, harvested by trypsinization, and fixed in 100% methanol at room temperature for 20 min. The cells were spread on slides, stained with DAPI solution (2 µg/mL), and analyzed under a FluoView confocal fluorescent microscope (FluoviewFV10i; Olympus Corporation, Tokyo, Japan).

### 2.7. Annexin V-PE Binding Assay

Apoptotic cell distribution was assessed using the MuseTM Annexin V and Dead Cell kit (MCH100105; Merck Millipore Co., Darmstadt, Germany) as per the manufacturer’s guidelines. The kit utilizes a fluorescent dye, phycoerythrin (PE), conjugated to Annexin V to detect phosphatidylserine externalization during apoptosis, along with 7-AAD (7-amino-actinomycin D) to identify dead cells. In brief, cells detached by trypsinization were collected via centrifugation at 500× *g* for 7 min at 4 °C, resuspended with MuseTM Annexin V and Dead Cell reagent, and analyzed using a Muse Cell Analyzer (0500-3115; Merck Millipore Co.).

### 2.8. Statistical Analysis

Experimental data are expressed as means ± standard deviations. The Mann–Whitney U test was performed to assess differences in non-normally distributed variables. Quantitative densitometry of Western blot bands, based on ImageJ analysis, was evaluated using SPSS 17.0 (IBM Corp., Armonk, NY, USA). Statistical significance was determined as *p* < 0.05.

## 3. Results

### 3.1. Expression of p-GSK3β in Melanoma Tissues and Normal Skin Tissues

Western blot analysis was conducted to assess the expression levels of p-GSK3β proteins in melanoma tissues (*n* = 6) and normal skin tissues (*n* = 6). The levels of p-GSK3β were found to be significantly elevated in all melanoma tissue samples relative to normal tissues (Figure 1A). The Mann–Whitney U test indicated a median (interquartile range) of 0.3104 (0.1386~0.7502) for normal skin tissues and 1.1951 (0.9500~2.0602) for melanoma tissues with respect to p-GSK3β expression, with a statistically significant difference (*p* = 0.0009) between the two groups (Figure 1B).

### 3.2. Cytotoxic Effects of Caffeic Acid Treatment on Human Melanoma Cells

To evaluate the anti-proliferative potential of caffeic acid (CA) (Figure 2A), G361 and SK-MEL-24 melanoma cells were treated with increasing concentrations of CA (0, 80, 160, 240, and 320 μM) for 24, 48, and 72 h. CA induced a dose- and time-dependent decrease in cell viability, with G361 cells exhibiting greater sensitivity than SK-MEL-24 cells. The IC_50_ values for G361 cells were 308.4 μM at 24 h, 172.7 μM at 48 h, and 144.8 μM at 72 h, indicating enhanced cytotoxicity over time (Figure 2B). ATP levels were measured following 24-h CA exposure, demonstrating that a concentration-dependent reduction in ATP levels was observed, particularly in G361 cells, suggesting mitochondrial dysfunction associated with cytotoxic stress (Figure 2C).

Morphological changes were also evident under phase contrast microscopy. Untreated G361 cells showed typical adherent morphology, whereas CA-treated cells exhibited cell shrinkage, detachment, and membrane blebbing in a concentration-dependent manner (Figure 2D). These alterations were consistent with apoptotic features, which were further confirmed by DAPI nuclear staining. CA treatment resulted in chromatin condensation and nuclear fragmentation, with apoptotic nuclei increasing at higher concentrations (Figure 2E). The degree of these structural changes was directly correlated with CA concentration, with higher doses resulting in more extensive morphological damage and detachment, in alignment with the reduction in cell viability.

To quantify apoptosis, Annexin V/PI staining followed by flow cytometry was performed (Figure 2F). The proportion of apoptotic cells increased significantly in a dose-dependent manner, with late apoptosis being the predominant form at higher CA concentrations.

### 3.3. Pro-Apoptotic and Cell Cycle Regulatory Effects of Caffeic Acid in G361 Cells via GSK3β Signaling

To further clarify the mechanisms underlying CA-induced cytotoxicity, we analyzed apoptosis induction, cell cycle progression, and p-GSK3β expression in G361 cells subjected to different CA concentrations. Western blot results indicated a marked increase in the protein levels of cleaved caspase-3, Bax, and cleaved PARP following CA treatment, paralleled by a concentration-dependent decrease in anti-apoptotic Bcl-2 (Figure 3A). As CA concentrations increased, the Bax/Bcl-2 ratio also rose, suggesting enhanced pro-apoptotic signaling. The Western blot assessment of regulatory proteins further showed CA-induced upregulation of p-GSK3β, p53, and p21, alongside reduced cyclin D1 expression, with effects more pronounced at higher concentrations (Figure 3B).

## 4. Discussion

GSK3β protein is recognized for its essential involvement in a range of cellular activities, such as metabolic regulation, cell signaling, cell cycle control, cell survival, and apoptosis [5]. Due to its role in these diverse processes, GSK3β may function as either a tumor suppressor or promoter, with its effect varying based on the specific context and cancer type [9]. The literature has extensively documented the multifaceted and frequently paradoxical roles of GSK3β in different cancer cell types [25]. As an illustration, GSK3β has been reported to facilitate tumor progression in osteosarcoma and pancreatic cancer, while acting to suppress cancer development in colorectal cancer cells [13,26,27].

In this study, we observed upregulation of p-GSK3β in melanoma tissue samples ex vivo, as demonstrated by Western blot analysis (Figure 1A,B). Previous research indicates that inhibition of GSK3β can promote p53 accumulation and induce apoptosis, supporting an anti-apoptotic function of GSK3β in malignant cells [28]. Our results align with findings and suggest that increased p-GSK3β (inactive form) expression in melanoma may contribute to tumor proliferation and progression.

The precise involvement of GSK3β in melanoma pathogenesis remains less explored in comparison to its established roles in other malignancies. Emerging evidence indicates that GSK3β signaling fosters melanoma oncogenicity by modulating both N-cadherin expression and the dynamics of focal adhesion complexes [29,30]. Notably, the inactivation of GSK3β by Ser9 phosphorylation appears especially relevant for melanoma progression, as our findings reveal a significantly higher expression of p-GSK3β in melanoma specimens versus normal skin tissues. This observation aligns with data from other cancer types in which elevated p-GSK3β is associated with increased tumor cell survival and resistance to apoptosis [18]. Inactivation of GSK3β through phosphorylation impairs its tumor suppressor roles, such as the regulation of β-catenin turnover and the control of cell cycle checkpoints, thus facilitating melanoma cell proliferation and promoting cell survival [31].

Caffeic acid is recognized as a multifunctional bioactive agent exhibiting a spectrum of anticancer effects that are not limited to its antioxidative properties. Earlier studies have shown that CA can profoundly alter essential oncogenic processes, such as suppressing proliferation, triggering apoptosis, and causing cell cycle arrest across several cancer cell types [21]. The documented pro-apoptotic activity of this compound extends to a range of malignancies, including cancers of the lung, colorectal, breast, and prostate origin [23,32]. On a mechanistic level, CA confers robust cellular defense against oxidative stress through its reactive oxygen species-scavenging capacity and the modulation of multiple redox-sensitive signaling pathways that are frequently altered in neoplastic cells [33].

Given CA’s well-established antioxidant properties and our findings regarding its impact on GSK3β phosphorylation, the connection between oxidative stress and GSK3β regulation is particularly pertinent for understanding the anticancer activity of CA. Prior research has shown that oxidative stress can modulate GSK3β activity, with reactive oxygen species frequently leading to GSK3β activation and promoting oncogenic pathways [34]. In contrast, antioxidant agents may promote inactivation of GSK3β by enhancing Ser9 phosphorylation [35]. Our data indicate that CA’s anti-melanoma efficacy could be attributed to this dual mechanism, whereby its antioxidant effects lower oxidative stress while concurrently facilitating GSK3β inactivation through phosphorylation. This mechanism aligns with our observed increases in p-GSK3β levels, the induction of apoptosis, and the normalization of cell cycle processes following CA administration, supporting the conclusion that CA can modulate the oxidative stress–GSK3β axis, which is often disrupted in melanoma.

In this study, we identified a dose-dependent increase in GSK3β Ser9-phosphorylation in response to CA exposure. This mechanism resembles the action of lithium chloride, a recognized and widely used clinical GSK3β inhibitor, implying that CA may act as a natural GSK3β modulator with potential therapeutic relevance [36]. Notably, the increase in p-GSK3β after CA treatment appears contradictory, given that the constitutive overexpression of p-GSK3β has been reported in melanoma tissues. These findings are paradoxical, suggesting that the efficacy of CA in inhibiting GSK3β signaling in melanoma cells may not be directly related to the induction of apoptosis. Although our findings strongly support CA’s therapeutic potential in melanoma via GSK3β modulation, this study possesses certain limitations that must be addressed. First, the small sample size of clinical tissues (*n* = 6) and the exclusive use of the G361 melanoma cell line may hinder the broader applicability of the results. Second, although we showed that CA influences GSK3β phosphorylation and its downstream pathways, the direct molecular interaction between CA and GSK3β was not verified by binding assays, leaving the precise mechanism unresolved for future investigation. Third, various methods, including GSK3β overexpression and knockdown assays, are needed to elucidate the biological consequences of p-GSK3β upregulation in melanoma tissues.

## 5. Conclusions

In conclusion, this study confirmed a significant increase in p-GSK3β expression in melanoma tissues and observed dose-dependent effects of CA on proteins related to the cell cycle and apoptosis, including activated tumor suppressor pathways (p53 and p21) and reduced cyclin D1 levels. CA treatment promoted GSK3β phosphorylation but did not appear to directly affect apoptosis or the cell cycle. Although further investigation is needed, the data from this study provide substantial preclinical evidence supporting the potential of CA as an anti-melanoma therapeutic candidate, as a natural compound possessing both GSK3β-modulating and antioxidant properties.

## Figures and Tables

**Figure 1 biomedicines-13-02389-f001:**
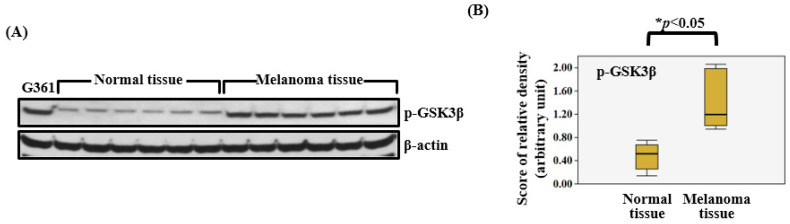
p-GSK3β protein expression in human normal skin and malignant melanoma tissues, and relative protein expression. (**A**) Western blot analysis. Levels of p-GSK3β were markedly increased in melanoma tissues compared to normal tissues. β-actin served as a loading control. (**B**) The medians of normal skin and melanoma tissues were statistically compared using the Mann–Whitney U test (*n* = 12, * *p* < 0.05).

**Figure 2 biomedicines-13-02389-f002:**
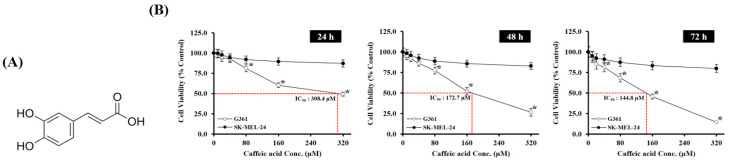
Cytotoxic effects of CA on G361 cells. (**A**) Chemical structure of caffeic acid. (**B**) G361 and SK-MEL-24 cells were treated with various concentrations of CA (0, 80, 160, 240, and 320 μM) for 24, 48, and 72 h. Cell viability was measured using the MTT assay. IC_50_ values for G361 cells were calculated for each time point. * *p* < 0.05 versus control. (**C**) Intracellular ATP levels were measured after treatment with CA (0, 20, 40, and 80 μM) for 24 h. (**D**) Phase contrast microscopy images of G361 cells treated with CA (0, 20, 40, and 80 μM) for 24 h. Scale bar: 50 μm. (**E**) Nuclear morphology was assessed using DAPI staining after 24 h of CA treatment. Apoptotic nuclear condensation and fragmentation were observed in a dose-dependent manner. Scale bar: 25 μm. (**F**) Quantification of apoptotic cells using the Annexin V-PE binding assay and Muse Cell Analyzer after 24 h of CA treatment.

**Figure 3 biomedicines-13-02389-f003:**
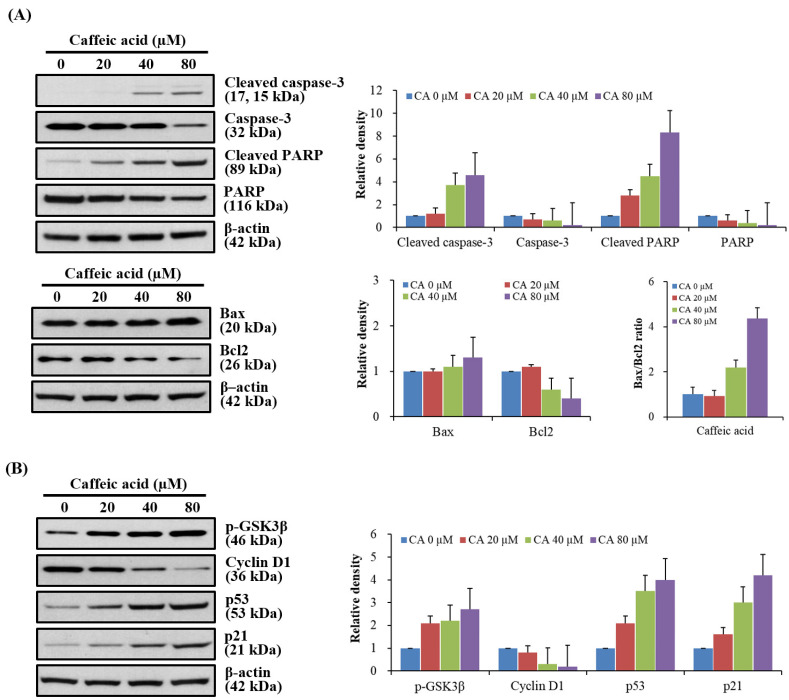
(**A**) Western blot analysis was used to evaluate pro- and anti-apoptotic protein expression. β-actin served as a loading control. The Bax/Bcl-2 ratio was measured after 48 h of incubation. (**B**) Caffeic acid induced dose-dependent changes in the expression of p-GSK3β and vital cell cycle regulators (Cyclin D1, p53, and p21) in melanoma cells, as detected by Western blot analysis. A quantitative analysis of each Western blot protein expression was performed, and the corresponding graphs are presented alongside the Western blot results. Error bars represent the mean ± SEM for three independent experiments.

**Table 1 biomedicines-13-02389-t001:** Dysregulation of GSK3β in various cancers.

Malignancy Type	Overexpression	Reference
Colorectal cancer	GSK3β	[13]
Pancreatic cancer	GSK3β	[14]
Hepatocellular carcinoma	GSK3β	[15]
Ovarian cancer	GSK3β	[16]
Renal cell carcinoma	GSK3β	[17]
Glioma	p-GSK3β (ser9)	[18]
Oral SCC	p-GSK3β (ser9)	[12]

## Data Availability

Data are included within the article.

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
