# Peer review of "Apoptosis, Cell Growth, and Glycogen Synthase Kinase 3β Phosphorylation in Caffeic Acid-Treated Human Malignant Melanoma Cells"

_biomedicines, 2025, doi:10.3390/biomedicines13102389_

Round 1
Reviewer 1 Report
Comments and Suggestions for Authors
In this manuscript, the authors investigate the expression of phosphorylated GSK3β (p-GSK3β) in human melanoma tissues and explore the effects of caffeic acid (CA) on melanoma progression. The study reports that p-GSK3β is significantly elevated in melanoma and that CA inhibits melanoma progression by modulating GSK3β activity. The objective is important and interesting; however, I have several major concerns that should be addressed before the manuscript can be considered further:
- The title of Figure 1 states “GSK3β protein expression in human…,” yet Figure 1A presents changes in p-GSK3β expression between normal and melanoma tissues. In addition, Methods section 2.3 indicates that GSK3β (SC-7291) was used for Western blot analysis. Please clarify whether the data represent total GSK3β or phosphorylated GSK3β.
- CA treatment was performed on G361 cells at different doses for 48 hours. Were additional treatment durations tested? If so, please provide data to evaluate potential time-dependent effects.
- The manuscript shows that CA treatment inhibits melanoma progression by inducing cytotoxicity and apoptosis in G361 cells. Since p-GSK3β is also implicated in promoting cancer cell migration and invasion, have the authors investigated whether CA treatment affects these processes?
- The authors only investigated the effects of CA treatment on G361 cell, can similar results be reproduced in other melanoma cell lines, such as A375 or SK-MEL-28?
- The images in Figure 2B are of low resolution. Please provide higher-quality images to ensure clarity and reproducibility.
- The original, unprocessed data for the Western blot analyses should be provided to ensure transparency and reproducibility.
Author Response
To Reviewer 1
For the reviewer's convenience, all revisions made in response to the comments have been highlighted in red text.
Comments and suggestions for Authors
[Comments No. 1]
The title of Figure 1 states “GSK3β protein expression in human…,” yet Figure 1A presents changes in p-GSK3β expression between normal and melanoma tissues. In addition, Methods section 2.3 indicates that GSK3β (SC-7291) was used for Western blot analysis. Please clarify whether the data represent total GSK3β or phosphorylated GSK3β.
[Response to Reviewer’s comments No. 1]
Lines 110-113 and Lines 188 : We agree with this comment. We apologize for the confusion in the Methods section. We actually used p-GSK3β (SC-3738) antibody for the experiments, not GSK3β (SC-7291) as initially stated. We have corrected this error in the revised manuscript. We chose to focus on p-GSK3β because there are limited studies investigating the relationship between the inactive form of GSK3β (p-GSK3β) and cancer, particularly in melanoma research.
[Comments No. 2]
CA treatment was performed on G361 cells at different doses for 48 hours. Were additional treatment durations tested? If so, please provide data to evaluate potential time-dependent effects.
[Response to Reviewer’s comments No. 2]
Lines 123-124 : Thank you for pointing this out. We have conducted additional experiments to evaluate time-dependent effects of CA treatment. We performed MTT assays on G361 and SK-MEL-28 cell lines treated with various concentrations of caffeic acid for 24, 48, and 72 hours to assess cell viability over different time points.
[Comments No. 3]
The manuscript shows that CA treatment inhibits melanoma progression by inducing cytotoxicity and apoptosis in G361 cells. Since p-GSK3β is also implicated in promoting cancer cell migration and invasion, have the authors investigated whether CA treatment affects these processes?
[Response to Reviewer’s comments No. 3]
Thank you for this valuable suggestion. Unfortunately, due to time constraints, we were unable to perform experiments examining whether CA treatment affects cancer cell migration and invasion by modulating p-GSK3β in the current study. Based on reviewer’s opinion, we plan to perform cell migration and invasion assays to provide more comprehensive mechanistic insights.
[Comments No. 4]
The authors only investigated the effects of CA treatment on G361 cell, can similar results be reproduced in other melanoma cell lines, such as A375 or SK-MEL-28?
[Response to Reviewer’s comments No. 4]
Lines 195-203 and Lines 221-225 : Agree. We have conducted additional experiments using the SK-MEL-24 cell line to evaluate the cytotoxic effects of caffeic acid treatment through MTT assay. SK-MEL-24 (amelanotic melanoma) is known to exhibit lower sensitivity to CAPE (caffeic acid phenethyl ester) compared to other melanoma cell lines. Even when CA demonstrates similar levels of cytotoxicity as CAPE, SK-MEL-24 cells may show relatively higher survival rates (i.e., lower toxicity) due to their inherent resistance characteristics. Both reference [1] and our current study consistently demonstrate that SK-MEL-24 cells exhibit less sensitivity to CA treatment compared to G361 cells. Due to this reduced sensitivity observed in SK-MEL-24 cells, we excluded this cell line from the apoptosis-related experiments to focus on the more responsive G361 model for mechanistic studies. The results have been incorporated into the revised manuscript with updated figures.
[Comments No. 5]
The images in Figure 2B are of low resolution. Please provide higher-quality images to ensure clarity and reproducibility.
[Response to Reviewer’s comments No. 5]
Lines 187, Lines 218-220 and Lines 280-282 : Agree. We have replaced all figures with higher-resolution images to ensure better clarity and reproducibility.
[Comments No. 6]
The original, unprocessed data for the Western blot analyses should be provided to ensure transparency and reproducibility.
[Response to Reviewer’s comments No. 6]
We have already provided the original, unprocessed data for all Western blot analyses to Editor.
This manuscript has not been submitted in whole or in part to any other journal, and its English language editing was completed by an English proofreading company (HARRISCO). The authors have no conflicts to report.
Some sentences that were vaguely explained have been rewritten to make them more clearly understandable, without changing their meaning. Also, grammatical problems and typos in the manuscript other than those mentioned by the reviewer have been corrected and marked in red.
Thank you for your valuable feedback. We tried to follow the Reviewer’s comments as much as possible. I apologize to you and reviewers for causing so much trouble with our manuscript.
References
[1] Hallan, S. S., Sguizzato, M., Mariani, P., Cortesi, R., Huang, N., Simelière, F., Marchetti, N., Drechsler, M., Ruzgas, T., & Esposito, E. (2020). Design and Characterization of Ethosomes for Transdermal Delivery of Caffeic Acid. Pharmaceutics, 12.

Reviewer 2 Report
Comments and Suggestions for Authors
This manuscript investigates the anti-melanoma effects of caffeic acid (CA) with a focus on its modulation of the GSK3β signaling pathway. The study demonstrates that p-GSK3β levels are elevated in melanoma tissues and that CA treatment suppresses cell proliferation, induces apoptosis, and modulates cell cycle regulators in G361 melanoma cells. The topic is interesting and potentially valuable, but the current version suffers from experimental limitations, mechanistic inconsistencies, and presentation issues that weaken the strength of the conclusions. Substantial revision is required before the manuscript can be considered for publication.
Major Concerns
-
Only one melanoma cell line (G361) was used throughout the experiments, which greatly limits the generalizability of the findings. Validation in at least one or two additional melanoma cell lines is strongly recommended.
-
The results present a mechanistic paradox: p-GSK3β is already upregulated in melanoma tissues, yet further elevation of p-GSK3β by CA is associated with apoptosis and growth suppression. This contradiction is insufficiently explained and requires a clearer mechanistic rationale to reconcile these observations.
-
No direct evidence is provided to demonstrate the interaction between CA and GSK3β. Without molecular binding assays, kinase activity assays, or genetic modulation (knockdown/overexpression), the causal link remains speculative.
-
Positive controls, such as treatment with known GSK3β inhibitors (e.g., lithium chloride), are missing. Including such controls would strengthen the claim that CA’s effects are specifically mediated through the GSK3β pathway.
-
Quantitative data presentation is incomplete. Several Western blot results lack proper densitometric analysis, and statistical reporting is inconsistent. Annexin V assay results also show relatively modest increases in apoptosis (up to ~21%), raising questions about the biological significance of the observed effects.
Minor Concerns
-
The Introduction is overly lengthy and somewhat repetitive, especially in summarizing CA’s effects in other cancers. This section could be streamlined to focus more directly on the gap in melanoma research.
-
Figures, particularly the Western blots, are of low resolution and should be improved with clearer band visualization and inclusion of molecular weight markers. Figure legends should also indicate the number of experimental replicates.
-
The Discussion section devotes extensive space to reviewing contradictory roles of GSK3β in other cancer types but does not clearly highlight the novelty or limitations of the present study. The narrative should be tightened to better emphasize this study’s contributions.
-
Methodological details require clarification. For example, the description of the MTT assay includes “siRNA treatment,” which appears inconsistent and may be an error. In addition, the statistical methods switch between Student’s t-test and the Mann–Whitney U test without justification; a normality test should be reported to support the choice of statistical analysis.
Author Response
To Reviewer 2
For the reviewer's convenience, all revisions made in response to the comments have been highlighted in red text.
Comments and suggestions for Authors
Major Concerns
[Comments No. 1]
Only one melanoma cell line (G361) was used throughout the experiments, which greatly limits the generalizability of the findings. Validation in at least one or two additional melanoma cell lines is strongly recommended.
[Response to Reviewer’s comments No. 1]
Lines 195-203 and Lines 221-225 : Thank you for this important suggestion. We have conducted additional experiments using the SK-MEL-24 cell line to evaluate the cytotoxic effects of caffeic acid treatment through MTT assay. SK-MEL-24 (amelanotic melanoma) is known to exhibit lower sensitivity to CAPE (caffeic acid phenethyl ester) compared to other melanoma cell lines. Even when CA demonstrates similar levels of cytotoxicity as CAPE, SK-MEL-24 cells may show relatively higher survival rates (i.e., lower toxicity) due to their inherent resistance characteristics. Both reference [1] and our current study consistently demonstrate that SK-MEL-24 cells exhibit less sensitivity to CA treatment compared to G361 cells. Due to this reduced sensitivity observed in SK-MEL-24 cells, we excluded this cell line from the apoptosis-related experiments to focus on the more responsive G361 model for mechanistic studies. The results have been incorporated into the revised manuscript with updated figures.
[Comments No. 2]
The results present a mechanistic paradox: p-GSK3β is already upregulated in melanoma tissues, yet further elevation of p-GSK3β by CA is associated with apoptosis and growth suppression. This contradiction is insufficiently explained and requires a clearer mechanistic rationale to reconcile these observations.
[Response to Reviewer’s comments No. 2]
We appreciate the reviewer's insightful observation regarding the apparent mechanistic paradox. In our study, we confirmed that p-GSK3β is upregulated in melanoma tissues. Based on this finding, when CA, which exhibits cytotoxic effects on melanoma cells, is applied to melanoma cell lines, p-GSK3β (Ser9) levels would generally be expected to decrease or undergo pathway-dependent changes. However, direct measurements of p-GSK3β levels in SK-MEL-24 or G361 cells following CA treatment have not been previously reported in the literature. Mechanistically, there exists substantial evidence that CA primarily inhibits PI3K/AKT signaling, thereby reducing GSK3β Ser9 phosphorylation and promoting GSK3β activation [2,3]. However, several literature reports have documented exceptions where p-GSK3β levels may paradoxically increase depending on various factors including melanoma subtype, treatment concentration and duration, cellular redox status, and activation of compensatory pathways such as p38 and MAPK signaling [4,5].
We plan to address these specific aspects through comprehensive follow-up studies currently in preparation.
[Comments No. 3]
No direct evidence is provided to demonstrate the interaction between CA and GSK3β. Without molecular binding assays, kinase activity assays, or genetic modulation (knockdown/overexpression), the causal link remains speculative.
[Response to Reviewer’s comments No. 3]
We agree with this comment. Due to time constraints, we were unable to perform siRNA knockdown, rescue assays, or molecular binding studies in the current study. We acknowledge that these experiments are essential to establish a definitive causal relationship between CA treatment and GSK3β modulation. We plan to address this limitation in our ongoing follow-up studies.
[Comments No. 4]
Positive controls, such as treatment with known GSK3β inhibitors (e.g., lithium chloride), are missing. Including such controls would strengthen the claim that CA’s effects are specifically mediated through the GSK3β pathway.
[Response to Reviewer’s comments No. 4]
Supplementary material : Agree. We have conducted additional experiments using lithium chloride (LiCl), a well-known GSK3β inhibitor, at various concentrations in G361 cells. Western blot analysis was performed to evaluate p-GSK3β expression levels following LiCl treatment. These results have been included as supplementary material, demonstrating similar p-GSK3β expression patterns compared to caffeic acid treatment.
[Comments No. 5]
Quantitative data presentation is incomplete. Several Western blot results lack proper densitometric analysis, and statistical reporting is inconsistent. Annexin V assay results also show relatively modest increases in apoptosis (up to ~21%), raising questions about the biological significance of the observed effects.
[Response to Reviewer’s comments No. 5]
Lines 280-282 and Lines 286-289 : We thank the reviewer for this important feedback. We have performed densitometric analysis for all Western blot results (Fig 3A, 3B) using ImageJ software. The quantitative graphs with error bars have been included in the revised figures.
Lines 154-159, Lines 204-216 and Lines 225-229 : We have conducted additional apoptosis experiments using 4′,6-diamidino-2-phenylindole (DAPI) staining(Fig 2E), which confirmed apoptotic nuclear changes at higher CA concentrations. Regarding the modest apoptotic effect observed with CA treatment, we plan to perform follow-up studies to better understand the clinical relevance of these effects.
Minor Concerns
[Comments No. 1]
The Introduction is overly lengthy and somewhat repetitive, especially in summarizing CA’s effects in other cancers. This section could be streamlined to focus more directly on the gap in melanoma research.
[Response to Reviewer’s comments No. 1]
Agree. In accordance with the reviewer’s comments, The Introduction has been streamlined to focus on the essential background information directly relevant to our melanoma research.
[Comments No. 2]
Figures, particularly the Western blots, are of low resolution and should be improved with clearer band visualization and inclusion of molecular weight markers. Figure legends should also indicate the number of experimental replicates.
[Response to Reviewer’s comments No. 2]
Lines 280-282 and Lines 286-289 : We appreciate the reviewer's feedback regarding figure quality and presentation. All Western blot experiments were performed in triplicate (n=3), and the corresponding error bars have been displayed in the figures. The figure legends have been updated to clearly indicate the number of experimental replicates for each experiment.
[Comments No. 3]
The Discussion section devotes extensive space to reviewing contradictory roles of GSK3β in other cancer types but does not clearly highlight the novelty or limitations of the present study. The narrative should be tightened to better emphasize this study’s contributions.
[Response to Reviewer’s comments No. 3]
Agree. The Discussion section has been revised to emphasize the specific contributions and novelty of our study.
[Comments No. 4]
Methodological details require clarification. For example, the description of the MTT assay includes “siRNA treatment,” which appears inconsistent and may be an error. In addition, the statistical methods switch between Student’s t-test and the Mann–Whitney U test without justification; a normality test should be reported to support the choice of statistical analysis.
[Response to Reviewer’s comments No. 4]
Lines 184-185 : Agree. Based on reviewer’s opinion, the methodological descriptions have been revised for clarity and accuracy.
This manuscript has not been submitted in whole or in part to any other journal, and its English language editing was completed by an English proofreading company (HARRISCO). The authors have no conflicts to report.
Some sentences that were vaguely explained have been rewritten to make them more clearly understandable, without changing their meaning. Also, grammatical problems and typos in the manuscript other than those mentioned by the reviewer have been corrected and marked in red.
Thank you for your valuable feedback. We tried to follow the Reviewer’s comments as much as possible. I apologize to you and reviewers for causing so much trouble with our manuscript.
References
[1] Hallan, S. S., Sguizzato, M., Mariani, P., Cortesi, R., Huang, N., Simelière, F., Marchetti, N., Drechsler, M., Ruzgas, T., & Esposito, E. (2020). Design and Characterization of Ethosomes for Transdermal Delivery of Caffeic Acid. Pharmaceutics, 12.
[2]Wang, R., Ruvolo, V., Jacamo, R., McQueen, T., Borthakur, G., Qiu, Y., Coombes, K., Zhang, N., Shpall, E., Champlin, R., Garzon, R., Marcucci, G., Croce, C., Konopleva, M., Andreeff, M., & Kornblau, S. (2013). Phosphorylation Of GSK3β Is Associated With Inferior Survival In Acute Myeloid Leukemia and Is An Indicator Of AKT Activation In AML Blasts and Bone Marrow Mesenchymal Stem Cells. Blood, 122, 2551-2551.
[3]Chen, Y., Favata, M., Pusey, M., Li, J., Lo, Y., Ye, M., Wynn, R., Wang, X., Yao, W., & Chen, Y. (2020). Identification of pAKT as a pharmacodynamic marker for MER kinase in human melanoma G361 cells. Biomarker Research, 8.
[4]Kadoma, Y., & Fujisawa, S. (2008). A Comparative Study of the Radical-scavenging Activity of the Phenolcarboxylic Acids Caffeic Acid, p-Coumaric Acid, Chlorogenic Acid and Ferulic Acid, With or Without 2-Mercaptoethanol, a Thiol, Using the Induction Period Method. Molecules, 13, 2488-2499.
[5]Hosseini, R., Moosavi, F., Silva, T., Rajaian, H., Hosseini, S., Bina, S., Saso, L., Miri, R., Borges, F., & Firuzi, O. (2018). Modulation of ERK1/2 and Akt Pathways Involved in the Neurotrophic Action of Caffeic Acid Alkyl Esters. Molecules, 23.

Round 2
Reviewer 1 Report
Comments and Suggestions for Authors
This is an interesting study and the authors used multivariate experimental design methodologies to collect a unique dataset. The paper has certain novelty and advantages for this field research work. Furthermore, the authors has addressed all the reviewer’s concerns. I suggest this manuscript can be published in “Biomedicines”.
Author Response
We sincerely appreciate your positive evaluation of our work and your thoughtful comments. We are grateful for your recognition of the novelty and significance of our study, and we are pleased that our revisions have addressed your concerns.
Reviewer 2 Report
Comments and Suggestions for Authors
1. The study analyzed patient tissues and reported that p-GSK3β levels were elevated in melanoma compared to normal skin, which is an interesting finding. However, this evidence alone is not sufficient to establish GSK3β as a therapeutic target in melanoma.
2. The manuscript is entitled “Caffeic Acid Suppresses Proliferation and Development of Human Malignant Melanoma Cells by Modulating Glycogen Synthase Kinase 3β”. However, the presented evidence merely demonstrates that caffeic acid promotes GSK3β phosphorylation. This does not prove that caffeic acid suppresses melanoma via GSK3β.
3. While caffeic acid induces phosphorylation of GSK3β, Figure 1 shows that p-GSK3β is already elevated in melanoma tissues compared to normal tissues. The biological consequence of this elevation is not demonstrated or explained.
4. The experimental design is rather simple, with limited data. The conclusions are not reliable, and the logical connections are weak.
Author Response
[Reviewer’s comment 1]
The study analyzed patient tissues and reported that p-GSK3β levels were elevated in melanoma compared to normal skin, which is an interesting finding. However, this evidence alone is not sufficient to establish GSK3β as a therapeutic target in melanoma.
[Response to Reviewer’s comment 1]
(1) Although the constitutive overexpression of p-GSK3β in melanoma tissues was a very interesting result, the finding that p-GSK3β is upregulated after CA treatment suggests that the apoptosis-inducing effect of CA in melanoma cells may not be directly related to the inhibition of GSK3β signaling.
(2) We revised and changed sentences in the Abstract, Discussion, and Conclusion sections based on reviewer's comment (page 1, lines 27-29; page 9, lines 302-306, 313-315, and 318-325).
[Reviewer’s comment 2] The manuscript is entitled “Caffeic Acid Suppresses Proliferation and Development of Human Malignant Melanoma Cells by Modulating Glycogen Synthase Kinase 3β”. However, the presented evidence merely demonstrates that caffeic acid promotes GSK3β phosphorylation. This does not prove that caffeic acid suppresses melanoma via GSK3β.
[Response to Reviewer’s comment 2]
(1) Current studies have confirmed that the increased sensitivity of melanoma cells to caffeic acid involves tumor suppressors p53, Bax, p21, and cyclin D1, but the involvement of GSK3β has not been clearly proven.
(2) We revised and added sentences in the Abstract, Discussion, and Conclusion sections based on reviewer's comment (page 1, lines 27-29; page 9, lines 302-306, 313-315, and 318-325).
(3) The title has also been changed in accordance with the revised conclusion (page 1, lines 2-4).
[Reviewer’s comment 3] While caffeic acid induces phosphorylation of GSK3β, Figure 1 shows that p-GSK3β is already elevated in melanoma tissues compared to normal tissues. The biological consequence of this elevation is not demonstrated or explained.
[Response to Reviewer’s comment 3]
(1) Our results suggest that the efficacy of CA in inhibiting GSK3β signaling (p-GSK3β upregulation) in melanoma cells may not be directly related to the induction of apoptosis, and we are preparing more experiments to interpret the biological consequences of p-GSK3β upregulation in melanoma tissue.
(2) We revised and added sentences in the Abstract, Discussion, and Conclusion sections based on reviewer's comment (page 1, lines 27-29; page 9, lines 302-306, 313-315, and 318-325).
[Reviewer’s comment 4] The experimental design is rather simple, with limited data. The conclusions are not reliable, and the logical connections are weak.
[Response to Reviewer’s comment 4]
(1) This study is the first to report, as a short communication rather than an original article, the upregulation of GSK3β phosphorylation in melanoma tissues and the modulation of its levels by caffeic acid in melanoma cells. Therefore, we regret that the experimental details are simple, the results are limited, and the significance of p-GSK3β upregulation is not clearly explained.
(2) The conclusion has been revised based on the author's comments, and any remaining unresolved issues will be reported through follow-up experiments (page 1, lines 27-29; page 9, lines 318-325).

Round 3
Reviewer 2 Report
Comments and Suggestions for Authors
There is no logical connection between Figure 1 and the overall theme of the study. Figure 1 mainly describes the relationship between p-GSK3β and melanoma, whereas the main focus of the study is the effect of caffeic acid on melanoma and its possible molecular mechanisms. Whether to accept the manuscript depends on the editor.